# Controlling Upper Limb Prostheses Using Sonomyography (SMG): A Review

**DOI:** 10.3390/s23041885

**Published:** 2023-02-08

**Authors:** Vaheh Nazari, Yong-Ping Zheng

**Affiliations:** 1Department of Biomedical Engineering, The Hong Kong Polytechnic University, Hong Kong SAR, China; 2Research Institute for Smart Ageing, The Hong Kong Polytechnic University, Hong Kong SAR, China

**Keywords:** controlling system, human–machine interface, machine learning, non-invasive sensor, prosthesis, sonomyography

## Abstract

This paper presents a critical review and comparison of the results of recently published studies in the fields of human–machine interface and the use of sonomyography (SMG) for the control of upper limb prothesis. For this review paper, a combination of the keywords “Human Machine Interface”, “Sonomyography”, “Ultrasound”, “Upper Limb Prosthesis”, “Artificial Intelligence”, and “Non-Invasive Sensors” was used to search for articles on Google Scholar and PubMed. Sixty-one articles were found, of which fifty-nine were used in this review. For a comparison of the different ultrasound modes, feature extraction methods, and machine learning algorithms, 16 articles were used. Various modes of ultrasound devices for prosthetic control, various machine learning algorithms for classifying different hand gestures, and various feature extraction methods for increasing the accuracy of artificial intelligence used in their controlling systems are reviewed in this article. The results of the review article show that ultrasound sensing has the potential to be used as a viable human–machine interface in order to control bionic hands with multiple degrees of freedom. Moreover, different hand gestures can be classified by different machine learning algorithms trained with extracted features from collected data with an accuracy of around 95%.

## 1. Introduction

Human–machine interfaces (HMIs) and wearable technologies have sparked a great deal of interest in recent decades because they can be used for a wide range of applications, including immersive games [1], rehabilitation engineering [2,3,4,5], the automotive industry [6,7], tele-operation in space [8], and virtual reality [9]. Furthermore, an HMI is frequently employed in the development of various control systems in prostheses and exoskeletons. In contrast to the many advancements in mechanical design, there are still significant challenges in regard to HMIs at higher levels of the control hierarchy to overcome. There is a specific type of interface that may be utilized to predict patients’ volitional movement from their residual muscle contractions or neuroactivities [10,11]. However, detecting a user’s motion intention fast enough to coordinate with devices is an important issue that requires further study [12]. A range of sensing modalities have been used to regulate human–machine interfaces. Sensing technologies for HMIs have been developed in order to provide accurate and trustworthy information to assist in the understanding of movement intentions.

In order to control prostheses, the most often used approach is the use of biological signals, which may be recorded by a variety of sensors and electrodes by interfacing with either the peripheral nervous system (PNS) or the central nervous system (CNS) [13,14]. This technique is classified as either non-invasive, including surface electromyography (sEMG), electroencephalography (EEG), forcemyography (FMG), mechanomyography (MMG), magnetoencephalography (MEG), force sensitive resistance (FSR), and magnetomicrometry (MM), with the last one being presently developed in MIT [15], or invasive, including implanted electromyography (iEMG), myoelectric implantable recording arrays (MIRAs), electroneurography (ENG), electrocorticography (ECoG), brain–chip interfaces (BCHIs), and magnetomicrometry (MM) [16]. Among all of these techniques, sEMG is the most commonly used method for prosthesis control, which has been studied very extensively [17,18,19,20].

Recently, there has been a concentrated attempt to non-invasively monitor user intention and intuitively operate various degrees of freedom of cutting-edge prostheses. This endeavor has been ongoing during the last decade. Non-invasive techniques include placing electrodes on the skin of the scalp or skeletal muscles, and applying conductive gel to the electrodes and skin surface in order to improve the contact area and conductivity between the electrodes and skin surface [21]. However, in order to collect low-amplitude electrical impulses from skeletal muscles, bipolar electrodes are put on the skeletal muscles in order to record muscular activities. However, there is a difficulty with the non-invasive technique in that the data obtained by sensors may be substantially influenced by a variety of circumstances, including electrode placement and movement, perspiration, and even noise caused by the electronic devices. Moreover, these methods have poor spatial resolution due to the interferences between the signals generated by neighboring or overlapping muscles. Surface EMG is also unable to accurately record the activity of deep muscles, and as a result, it is difficult to utilize this approach to control protheses with multiple degrees of freedom [22]. Additionally, training users to control robots by using biological signals is difficult and requires time, which is another drawback of these interface methods [23], as the signals are often not linearly related to the muscle outputs, such as force or angle [22].

Biomaterials have been used for implants for a long time [24]. Implanted myoelectric sensors, peripheral nerve implants, targeted muscle reinnervation, brain–computer interfaces [25], and implanted stimulators [26] are examples of new technologies and methods that have the potential to provide significant improvements and new opportunities in neurological research. Invasive techniques include the placing of neural implants deep into the brain, on the nerves or the skeletal muscles [16]; and the recording of signals from the cerebral cortex, part of the brain, or muscle activity. These implants are able to connect with the brain, nerves, and muscles to collect electrical signals during nerve or muscle activation. In addition, they give electrical impulses to neurons, as well as transmit electrical signals between neurons and computers, or between computers and neurons through a chip [24,27]. While invasive approaches may increase the stability of biological signals, as well as give more accurate information about the activities of the brain or muscles [28], these novel interface methods raise a lot of concerns regarding the safety and efficacy of the operations, which involve surgery or implanted devices [23]. Furthermore, these signals also have the presence of noises, the same as non-invasive techniques.

Researchers have also made significant efforts in recent years to employ new technologies and propose novel techniques for controlling prosthetic hands, such as augmented-reality (AR) glasses [29], inductive tongue control systems (ITCSs) [30], voice commands, and inertial measurement units (IMUs) [31,32]. Some concepts have proved that even the simplest techniques may have compelling results.

These techniques are often utilized for prostheses that only have a single degree of freedom. Hence, the analysis or classification of biological signals necessitates the development of intelligent characteristic algorithms that are capable of accurately classifying the different signals gathered with the least number of errors [33]. Utilizing a variety of machine learning methods, including deep learning, significant improvements in the processing and classification of biological signals have been made in recent years. For example, the use of machine learning has yielded good results and achieved high performance accuracy across a wide variety of topics, including the rehabilitation and re-education of physically handicapped human limbs [34]. In enhancing robot control, various algorithms, such as K nearest neighbors (KNN), support vector machines (SVMs), principal component analysis (PCA), linear discriminant analysis (LDA), artificial neural networks (ANNs), convolutional neural networks (CNNs), and Bayes networks, can be used to classify signals with an accuracy of approximately 90% [35].

Recently, it has been proven that replacing biological signals with ultrasound (US) imaging that may provide real-time dynamic images of interior tissue movements linked with physical and physiological activity enables better discernment between discrete motions or categorization of full finger flexion [36]. Muscle architectural changes can be detected by putting an ultrasound probe on the residual limb and by classifying different hand gestures based on muscle movement and activities for controlling a prosthesis [37].

Biosensing approaches and novel wearable devices, such as the sonomyography (SMG) technique for the implementation of control for upper limb prostheses, as well as machine learning algorithms for hand gesture recognition, are reviewed in this paper. The objective of this paper is to provide information about SMG systems for controlling upper limb prostheses based on the sensing of architectural changes in a subject’s muscles during contraction. Section 2 describes, in detail, the history of the SMG approach for controlling prostheses throughout the years, different modes of US, feature extraction for increasing the accuracy of classification, artificial intelligence (AI), and innovative decoding methods for hand movement classification.

## 2. Methodology

Available articles on upper limb prostheses and different controlling and HMI methods, especially controlling robots using SMG, published between 2004 and 2022 were reviewed using Google Scholar and PubMed resources in English. For this review paper, the combination of the keywords “Human Machine Interface”, “Sonomyography”, “Ultrasound”, “Upper Limb Prosthesis”, “Artificial Intelligence”, and “Non-Invasive Sensors” was used to search for articles. Sixty-one articles were found, of which fifty-nine were used in this review, and the two discarded articles were found to not be relevant (Figure 1).

For the first time in 2006, the SMG method as a novel HMI technique was presented. In the past 16 years, different groups have tried to study the potential of US to be utilized in controlling upper limb prostheses. To review the different feature extraction methods and machine learning algorithms to control a robotic hand by using three distinct US modes and evaluate the progression of accuracy and reliability of SMG as a HMI method, 16 articles published by different groups were utilized.

The original research publications, as well as review articles published in English between the years 2004 and 2022, were considered for inclusion in this article. However, case reports, editorials, and commentaries were among the types of publications that did not meet the requirements to be reviewed in this article.

## 3. Sonomyography (SMG)

The use of ultrasonic technology in sensor implementation for identifying finger motions in prosthetic applications has been researched over the last ten years. A ground-breaking study by Zheng et al. investigated whether ultrasound imaging of the forearm might be used to control a powered prosthesis, and the term “sonomyography” (SMG) was coined by the group [38]. Ultrasound signals have recently garnered the interest of researchers in the area of HMIs because they can collect information from both superficial and deep muscles and so provide more comprehensive information than other techniques [39]. Due to the great spatiotemporal resolution and specificity of ultrasound measurements of muscle deformation, researchers have been able to infer fine volitional motor activities, such as finger motions and the dexterous control of robotic hands [40,41]. To retain performance, a prosthesis that responds to the user’s physiological signals must be fast to respond. EEG, sEMG, and other intuitive interfaces are capable of detecting neuromuscular signals prior to the beginning of motion; therefore, they are predicted to appear before the motion itself [42,43,44]. However, ultrasound imaging can detect skeletal muscle kinematic and kinetic characteristics [45], which indicate the continued creation of cross bridges during motor unit recruitment and prior to the generation of muscular force [43,46], and these changes occur during sarcomere shortening, when muscle force exceeds segment inertial forces, and before the beginning of joint motion [43]. It is important to note that the changes in kinetic and kinematic ultrasonography properties of muscles occur prior to joint motion. As a result, prosthetic hands will be able to respond more quickly in the present and future.

### 3.1. Ultrasound Modes Used in SMG

Real-time dynamic images of muscle activities can be provided by US imaging systems. There are five different types of ultrasound modes, and each of them generates different information, but only some of them are applicable for use in controlling artificial robotic hands. The most popular ultrasound modes utilized in prosthesis control are A-mode, B-mode, and M-mode.

(1) A-mode SMG: One of the most basic types of US is A-mode, which offers data in one dimension in the form of a graph in which the *y*-axis indicates information about echo amplitude and the *x*-axis represents time, similar to the way that EMG signals indicate muscle activity.

In 2008, Guo et al. [47] introduced a novel HMI method called one-dimensional sonomyography (1D SMG) as a viable alternative to EMG for assessing the muscle activities and controlling protheses. In this study, nine healthy volunteers were asked to perform different types of hand and wrist movements. During these experiments, different data were collected, such as joint angles, EMG signals of forearm muscles, and muscle activities collected from A-mode Ultrasound. The results of their study showed that the 1D SMG technique can be reliable and has the potential to be used for controlling one-degree-of-freedom bionic hands.

A study by Guo et al. [48] was carried out in order to assess and compare the performance of one-dimensional A-mode SMG and sEMG signals while following guided patterns of wrist extension. They also looked at the possibility of using the 1D SMG to control bionic hands. They invited 16 healthy right-handed participants to conduct a variety of wrist motions with a variety of guided waveforms at a variety of movement speeds for their experiment. During wrist motions, a 1D SMG transducer with a sEMG electrode was connected to the forearm of participants, making it possible for them to record and capture the activity of the participants’ forearm muscle groups. Root mean squares (RMSs) were computed from the extensor carpi radialis after normalizing the signals obtained from the SMG and sEMG after they had been collected and normalized, respectively. When comparing the abilities of SMG and sEMG to follow guiding waveform patterns, the paired t-test was utilized to make the comparison. In addition, one-way analysis of variance (ANOVA) was utilized to determine the differences in SMG performance at different movement speeds. For sinusoidal, square, and triangular guiding waveforms, the mean RMS tracking errors of SMG were found to be between 13.6% and 21.5%, whereas sEMG was found to be between 24% and 30.7%. The results of a paired t-test experiment revealed that the RMS errors of SMG tracking were much lower than those of sEMG tracking.

When Guo and her colleagues [49] successfully tested the A-mode US on healthy participants, they used the same procedure on an amputee (Figure 2A). Participants in the study were instructed to extend their phantom wrist in order to control the prosthetic hand. Her research found a correlation between muscle thickness and wrist extension angle with a correlation coefficient of 0.94. Furthermore, the relationship between wrist angle and muscle thickness was studied, and they calculated the mean ratio of angle deformation, which was around 0.13%.

As a continuous part of their research, Chen et al. [50] investigated whether it is feasible to control a prosthetic hand with one degree of freedom by using muscle thickness variations recorded by a one-degree-of-freedom SMG. With varying patterns and movement speeds, nine right-handed healthy individuals were instructed to operate a prosthetic hand with their wrist motions and match the visual input with the target track. The opening position of the prosthesis was controlled by SMG signals from the subject’s extensor carpi radialis muscle. A prosthesis opening position was measured using an electronic goniometer in this investigation. The tracking error between the opening position of the prosthetic hand and the target track was computed in order to evaluate the performance of the controlling system. This study’s findings indicated that the SMG control’s mean RMS tracking errors ranged from 9.6% to 19.4% while moving at various speeds.

**Figure 2 sensors-23-01885-f002:**
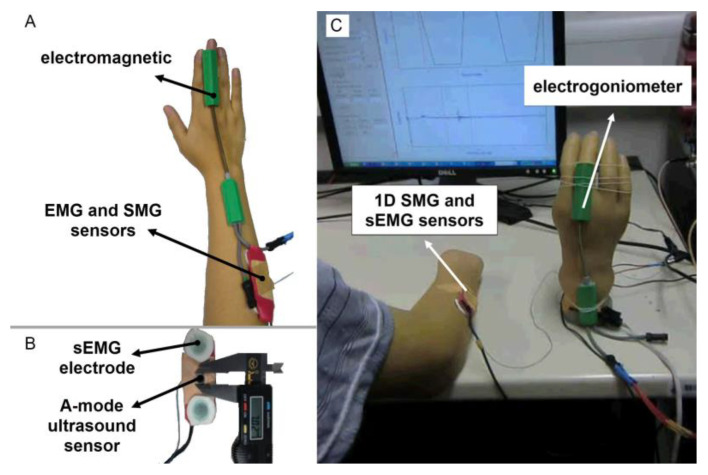
(**A**) The placement of the electro goniometer and sensors on healthy volunteers. (**B**) Placing A-mode small transducer (with a diameter of 7 mm) in between sEMG electrodes to collect both EMG and SMG signals from extensor carpi radialis muscle, simultaneously [51]. (**C**) The original image of the experimental setting, conducted by Guo and her colleagues in 2010. A-mode SMG setting for collecting SMG and EMG signals from a residual forearm for controlling a prosthesis to compare their performances, with the screen showing the A-mode ultrasound signal (lower half) and the guiding signal for muscle contraction (upper half).

In a study published in 2013, Guo et al. [51] further employed three different machine learning approaches to estimate the angle of the wrist, using a one-dimensional A-mode ultrasonic transducer, and the results were promising. During the experiment, nine healthy volunteers were instructed to execute wrist extension exercises at speeds of 15, 22.5, and 30 cycles per minute, while an A-mode ultrasound transducer recorded data from the participants’ forearm muscles (Figure 2B,C).

Because of the ability of US transducers to detect morphological changes in deep muscles and tendons, Yang et al. [52] presented a US-driven HMI as a viable alternative to sEMG for dexterous motion identification. Four A-mode piezoelectric ceramic transducers were built for their study. A custom-made armband was constructed to fix the four transducers while capturing the activity of the flexor digitorum superficialis (FDS), flexor digitorum profundus (FDP), flexor pollicis longus (FPL), extensor digitorum communis (EDC), and extensor pollicis longus (EPL), which all play a critical part in finger movements, including flexion and combined finger motions. Participants were asked to make 11 different hand gestures and hold such gestures for 3 to 5 s throughout the offline trial. Due to the fact that the raw echo signals obtained from the A-mode ultrasound transducer are constantly distorted by scattering noises and attenuation in tissues, signal processing was accomplished using temporal gain compensation (TGC), Gaussian filtering, Hilbert transform, and log compression [53].

In 2020, Yang et al. [54] suggested subclass discriminant analysis (SDA) and principal component analysis (PCA) to simultaneously predict wrist rotation (pronation/supination) and finger motions while using wearable 1D SMG system. They carried out trials both offline and online. In offline studies, eight tiny A-mode ultrasound transducers were mounted onto the hands of eight healthy volunteers, and the forearm muscles were captured using the transducers. In their study, the wrist rotations and eight kinds of finger motions (rest, fist, index point, fine pinch, tripod grasp, key grip, peace sign, and hang loose) were investigated. However, in the online test, a customized graphical user interface (GUI) was employed to conduct a tracking task in order to validate the simultaneous wrist and hand control. The results of this study showed that it was possible to classify the finger gestures and wrist rotation simultaneously while using the SDA machine learning algorithm, with an accuracy of around 99.89% and 95.2%, respectively.

In 2020, Engdahl et al. [55] proposed a unique wearable low-power SMG system for controlling a prosthetic hand. The proposed SMG system comprised four single-element transducers that were driven by a 7.4 V battery and operated at a constant frequency. In their investigation, a portable ultrasound transducer was fixed to the hands of five healthy participants in order to obtain muscle activity data. The data collected from participants were used to train an AI model in order to classify different finger movements. The results of this study showed that, using their proposed method, it was possible to classify nine different finger movements with an accuracy of around 95%.

(2) B-mode SMG: B-mode, or 2D mode, provides a cross-sectional image of tissues or organs and is one of the most popular US modes used in a wide range of medical applications. In B-mode US, organs and tissues show up as points of different brightness in 2D grayscale images made from the echoes. B-mode ultrasound can provide a real-time image of muscles under contraction.

Zheng et al. [38], for the first time, studied the potential of a portable B-mode ultrasound scanner for evaluation of the dimensional change of muscles and control of prosthetic hands. In their study, six healthy volunteers and three amputee participants were asked to perform wrist flexion and extension in order to capture the activities of forearm muscles (Figure 3). The morphological deformation of forearm muscles during activities was effectively identified and linearly linked with wrist angle. The mean ratio of wrist angle to percentage of forearm muscle contraction was evaluated in normal participants. When the three amputee participants engaged their residual forearm muscles, the SMG signals from their residual forearms were likewise recognized and recorded satisfactorily. They discovered that SMG may be used to regulate and monitor musculoskeletal disorders as a consequence of their research.

Shi et al. [57] employed B-mode ultrasound imaging to capture muscle activity during a finger’s flexion and extension. Artificial intelligence was then utilized to determine which fingers had been bent in various directions. All of the information was handled offline. A total of 750 sets of US pictures were obtained, with images from each group selected from forearm muscles during finger flexion and extension.

Ortenzi et al. [21] reported the use of ultrasound as a hand prosthesis HMI. Using a portable ultrasonic scanner equipped with a linear transducer, US pictures were captured and processed in the B-mode (2D imaging) in order to show the transverse section of the forearm underneath the transducer as a grayscale image. In the testing, the US transducer remained in position on the wrist thanks to an elastic band attached to a special plastic cradle. Specifically, this was performed in order to limit the amount of motion artefacts that would arise. Specifically, the goal of this research was to evaluate the categorization of ten various hand postures and grab forces.

Employing a computationally efficient approach to distinguish between complicated hand movements, Akhlaghi and colleagues [58] presented a real-time controlling system in relation to stroke rehabilitation and performed basic research into motor control biomechanics and artificial robotic limb control to analyze the feasibility of using 2D-mode US as a robust muscle computer interface and evaluate the possible therapeutic applications. They used a B-mode ultrasound transducer to evaluate the possibility of the classification of complex hand gestures and dexterous finger movements. In their study, dynamic ultrasound pictures of six healthy volunteers’ forearm muscles were provided, and these data were evaluated to map muscle activity based on the muscle deformation during diverse hand movements.

In 2017, McIntosh et al. [59] looked at how suitable different forearm mounting positions (transverse, longitudinal, diagonal, wrist, and posterior) were for a wearable ultrasound device. This is because the location of a device has a big impact on how comfortable it is and how well it works. In their study, in order to fix the B-mode US transducer on the participants’ arms, they designed a fixture manufactured by a 3D printer and strap. The gloves also had flexible sensors sewn into them so that they could measure the precise angle of each finger’s flexion.

In a 2019 study, Akhlaghi et al. [60] evaluated the impact of employing a sparse set of ultrasound scanlines in order to find the best location on the forearm for capturing the maximal deformation of the primary forearm muscles during finger motions, as well as classifying different types of hand gestures and finger movements. Five subjects were asked to make four different hand movements in order to see how the FDS, FDP, and FPL muscles worked.

In 2021, Fernandes et al. [61] developed a wearable HMI that made use of 2D ultrasonic sensors and non-focused ultrasound. The ultrasound radiofrequency (RF) signals were captured by using a B-mode linear array ultrasound probe while five healthy volunteers performed individual finger flexions. To intentionally diminish the lateral resolution of the ultrasound data, RF waves were averaged into fewer lateral columns. For full resolution, the first and third quartiles of classification accuracy were found to be between 80% and 92%. Using the suggested feature extraction approach with discrete wavelet transform, averaging into four RF signals might obtain a median classification accuracy of 87%. According to the results of their study, the authors mentioned that low-resolution images can have the same level of accuracy as high-resolution images.

(3) M-mode SMG: An M-mode scan, also known as a motion mode scan, uses a series of A-mode scan signals, normally by selecting one line in B-mode imaging, to depict tissue motion over time. Using the M-mode, it is possible to estimate the velocity of individual organ structures. In comparison to the B-mode and A-mode, the motion mode US scans at a greater frequency and provides more comprehensive information about the tissue.

Li et al. [39] conducted a study to determine the possibility of using M-mode ultrasound to detect wrist and finger movements. They compared M-mode and B-mode ultrasonography performance in the classification of 13 wrist and finger movements. A total of 13 movements were performed on eight healthy participants. Stable ultrasound data were collected by placing an ultrasound probe on an arm with a custom-made transducer holder. In order to cover the muscles of the forearm that are responsible for finger flexion and extension, the transducer was positioned at about halfway along the forearm’s length. During the same procedure, to ensure that the comparison was fair, the M-mode and B-mode ultrasound signals were both collected from the forearm. As a consequence of their investigation, M-mode SMG transducers were shown to be as accurate as B-mode SMG signals in detecting wrist and finger movements, as well as distinguishing between diverse hand gestures, and they may be employed in HMIs.

### 3.2. Muscle Location and Probe Fixation

It is vital to note that the position and location of the probe are critical in order to have greater control over robotic hands. The main muscles which perform different types of finger flexion are the FDS, FDP, and FPL muscles. However, to perform different wrist movements, the pronator teres, flexor carpi radialis, flexor carpi ulnaris, palmaris longus, and pronater quadratus are involved (Figure 4).

Hence, the placing of sensors to collect these muscle activities with better and more reliable control over the robot is important. After collecting data from healthy volunteers, Akhlaghi et al. [60] discovered that muscular distortion was significant in 30–50% of the forearm length from the elbow and that this region is the best place to record muscle movements for controlling robots. However, after testing various locations and fixing positions on a range of healthy individuals, McIntosh et al. [59] discovered that the wrist region is the most effective place for classifying discrete motions. Furthermore, they observed that the diagonal position is the most effective position for collecting data for identifying discrete gestures, whereas the diagonal and transverse positions are the most effective for predicting finger angles (Figure 5).

### 3.3. Feature Extraction Algorithm

To classify the finger movements and different hand gestures, it is important to use different types of algorithms to extract features from signals or images captured by US transducers because machine learning algorithms cannot process all the information. It is worth mentioning that using a machine learning algorithm without extracting features can classify different hand gestures, but the accuracy would be significantly less.

Shi et al. [57] captured the forearm muscle activities and controlled a hand prosthesis with B-mode ultrasound, and AI was used to classify the finger movements. Before using collected data to train their AI, a deformation field was constructed to extract features from the data after registering the ultrasound image pair with the demons registration algorithm for each group. Valerio Ortenzi [21] used the SMG technique as a valid HMI method to control a robotic hand. In order to classify ten different hand gestures and grasp forces, visual characteristics such as regions-of-interest gradients and histogram of oriented gradient (HOG) features were extracted from the collected images, and these features were used to train three machine learning algorithms.

The activity pattern was generated using an image-processing method developed by Akhlaghi et al. [58]. MATLAB (MathWorks, Natick, MA, USA) was used to extract the activity patterns for each kind of hand movement from the B-mode ultrasound picture frames. Pixel-wise differences were determined and then averaged across a time span to identify the spatial distribution of intensity variations that corresponded to the muscle activity in each sequential frame of each series (raw activity pattern). A hand motion was mapped to a single activity pattern by using this method. On the basis of the global thresholding level and decimal block size, the raw activity pattern was then transformed into a binary image. This database was then used to train the nearest neighbor classification algorithm.

McIntosh et al. [59] collected data from the forearm muscles of subjects in order to evaluate the effect of probe position on the control of a hand prosthesis. They utilized a B-mode US transducer to capture the muscle activities of volunteers. Before using the collected data to train their AI, the optical flow between the first frame of the new session and the base frame of the training set was estimated. The flow was then averaged to generate a 2D translation and to reduce mistakes caused by US displacement, which might result in differing anatomical characteristics. Following that, modification was made to the current video in order to better match the training and sample characteristics.

In a study conducted by Yang et al. [52], before using the collected data to train the machine learning model, the feature extraction process was carried out using segmentation and linear fitting to increase the accuracy of classification. Inspired by Castellini and colleagues [63,64], first-order spatial features were used to guide the feature extraction procedure. After selecting an evenly spaced grid of interest spots in the ultrasound picture, plane fitting was used to identify the spatial first-order features. Nevertheless, in their technique, the plane fitting was turned into linear fitting [65]. It was because of this change that the approach could be used for one-dimensional ultrasonic data.

In 2020, Yang et al. [54] classified and detected simultaneous wrist and finger movements, using SDA and PCA algorithms. To train their AI model, the characteristics of the data collected from participants were extracted using the Tree Bagger function, and the Random Forest method was used to evaluate the significance of characteristics. After that, two kinds of statistically significant characteristics were concatenated together for further analysis.

Fernandes et al. [61] used the LDA method to classify finger movements by using B-mode SMG. To make the classification more reliable and accurate, the authors used two different methodologies to extract characteristics from the data collected from volunteers. First, using the discrete wavelet transform (DWT) approach, the average RF signals were preprocessed prior to being used in the second method. In the next step, the mean absorption value (MAV) of the detail coefficient at various levels, as determined by the DWT approach, was determined. The second technique involves calculating a linear function over segmented portions of the envelope along the depth by using linear regression (LR). It was decided to utilize the slopes and intercepts of the predicted linear function as spatial characteristics in this study.

Li et al. [39] compared the productivity of B- and M-mode ultrasound transducers in relation to controlling an artificial robotic hand. In their study, they collected data from participants, and then the features from signals collected from an M-mode probe were extracted using a linear fitting approach, while the features from pictures captured with a B-mode transducer were extracted using a static ultrasound image method. These features were used for training the SVM algorithm.

### 3.4. Artificial Intelligence in Classification

To have dexterous and precise control over prostheses, different deep learning and machine learning algorithms have been developed to classify different hand gestures and intended movements, using SMG with high accuracy.

To control a prosthetic device in real time, Shi et al. [56] looked at the sum of absolute differences (SAD), the two-dimensional logarithmic search (TDL), the cross-correlation (CC) method, and algorithms such as SAD and TDL in conjunction with streaming single-instruction multiple-data extensions (SSEs). They utilized a block-matching method to measure the muscle deformation during contraction. To compare TDL with and without SSE, the findings revealed good execution efficiency, with a mean correlation coefficient of about 0.99, a mean standard root-mean-square error of less than 0.75, and a mean relative root-mean-square error of less than 8.0%. Tests have shown that a prosthetic hand can be controlled by only one muscle position, which allows for proprioception of muscle tension. They mentioned that SMG is good at controlling prosthetic hands, allowing them to open and close proportionally and quickly.

In order to capture muscle activity in a finger’s flexion and extension and evaluate the potential of using an ultrasound device in HMI, Shi et al. [57] employed B-mode ultrasound imaging. The deformation field was used to extract features, which were then inputted into the SVM classifier for the identification of finger movements. The experimental results revealed that the overall mean recognition accuracy was around 94%, indicating that this method has high accuracy and reliability. They assert that the suggested approach might be utilized in place of surface electromyography for determining which fingers move in distinct ways.

Guo and her colleagues [51] conducted a study and asked nine healthy volunteers to perform different wrist extensions; meanwhile, an A-mode portable probe was used to capture the activities of the extensor carpi radialis muscle. An SVM, a radial basis function artificial neural network (RBFANN), and a backpropagation artificial neural network (BP-ANN) were trained by data collected from extension exercises at 22.5 cycles per minute, and the rest of the data were used for cross-validation. For the purpose of evaluating the accuracy of the predictions made by the AI models utilized in their research, correlation coefficients and relative root-mean-square error (RMSE) were calculated. The findings revealed that the SVM method is the most accurate in predicting the wrist angle, with an RMSE of 13% and a correlation coefficient of 0.975%.

In 2015, Ortenzi et al. [21] proposed an advanced HMI method for using US devices. In their study, data were collected from three healthy participants, using B-mode ultrasound, in order to train a machine learning algorithm to classify different hand gestures. The first dataset included US pictures of six hand postures and four functional grasps, each with just one degree of grip force. The second dataset was used to evaluate the capacity to recognize various degrees of force for each kind of grip. In order to classify photos from the five datasets, an LDA classifier, a Naive Bayes classifier, and a decision tree classifier were used, among other methods. The LDA classifier trained with features extracted by HOGs outperformed the others and achieved 80% success in categorizing 10 postures/grasps and 60% success in classifying functional grasps with varied degrees of grip force in an experiment involving three intact human volunteers.

In order to classify complex hand gestures and dexterous finger movements, Akhlaghi et al. [58] collected the forearm muscle activities in different hand gestures in conjunction with wrist pronation. Using the activity patterns collected during the training phase, a database of potential hand movements was created, and the nearest neighbor classifier was used to categorize the various activity patterns using the database. The feature vectors in closest neighbor classification were created using two-dimensional activity pattern pictures, and the distance metric in a classification algorithm was determined by the cross-correlation coefficient between two patterns. For each participant, a database of activity patterns corresponding to various hand gestures was created during the training portion of the study. It was discovered that, during the testing phase, unique activity patterns were categorized using the database, with an average classification accuracy of 91%. A virtual hand could be controlled in real time by using an image-based control system that had an accuracy of 92% on average.

McIntosh et al. [59] collected data from participants’ forearm muscles in order to classify 10 different hand gestures, using US. In order to identify the finger positions or estimate finger angles, two machine learning algorithms were used. However, because machine learning algorithms cannot process all of the information, an optical flow was used to classify discrete gestures, and a first-order surface was used to detect finger angle. SVM and MLP algorithms were used to classify the different gestures and finger flexing in different joints. The results of this study showed that finger flexion and extension for performing 10 different hand gestures were classified after using image processing and neural networks with an accuracy of above 98%. They also found out that the MLP algorithm had a slight advantage over the SVM method in every location. After analyzing the data collected from finger flexion and extension in different joints, they mentioned that it is possible to classify the flexion and extension of each finger in different joints with an accuracy of 97.4%.

In an experiment reported by Yang et al. [52], in order to classify and identify the finger movements from using a wearable 1D SMG system, the muscle activity of participants during the performance of 11 different hand gestures was collected. Then the data were used to train LDA and SVM algorithms to classify hand movements. It was decided to use a five-fold cross-validation method. All the information was gathered in one database, which was then separated into five sections randomly and evenly distributed among them. One of the five components was designated as a testing set, while the other four were designated as training sets. The trial findings indicated that the accuracy of offline recognition was up to 98.83 ± 0.79%. The completion percentage of real-time motions was 95.4 ± 8.7%, and the time required to choose an online move was 0.243 ± 0.127 s.

In order to classify the finger movements, Akhlaghi et al. [60] used a B-mode ultrasound probe to capture the main forearm muscles’ activities. In addition, three different scanline reductions were used to limit the scanlines of the US. The data, after being collected and limited, were used to train a nearest neighbor algorithm to classify different finger movements and different hand gestures. Using the complete 128 scanline picture, the classification accuracy was 94.6%, while using four equally spaced scanlines, the classification accuracy averaged 94.5%. On the other hand, there was no significant difference in the ability to categorize items when the best scanlines were selected using fisher criteria (FC) and mutual information (MI). They also suggested that, instead of using the whole imaging array, a select subset of ultrasonic scanlines may be employed, which would not result in a reduction in classification accuracy for multiple degrees of freedom. Wearable sonomyography muscle–computer interfaces (MCIs) may also benefit from selecting a restricted number of transducer parts to decrease computation, instrumentation, and battery use.

To detect finger movements and wrist rotation simultaneously, Yang et al. [54] collected data from muscle activities during different finger movements with wrist rotation. Before using the collected data in the training of machine learning algorithms, different techniques were used to extract the features. The simultaneous wrist rotation and finger motions were predicted using an SDA technique and a PCA approach. The results indicated that SDA is capable of accurately classifying both finger movements and wrist rotations in the presence of dynamic wrist rotations. Using three subclasses to categorize wrist rotations, it is possible to properly classify around 99% of finger movements and 93% of wrist rotations. They also discovered that the wrist rotation angle is linearly related to the first principal component (PC1) of the chosen ultrasonography characteristics, independent of the finger motions being used. With just two minutes of user training, it was possible to achieve wrist tracking precision (R2) of 0.954 and finger-gesture categorization accuracy (96.5%) with the PC1.

Fernandes et al. [61] developed a wearable SMG technology to classify and categorize finger flexion and extension. In their study, 2D-mode US was used to collect five subjects’ muscle activities during finger movements. Before the LDA method was employed to categorize the finger motions, a feature selection process was carried out. The number of spatial and temporal characteristics that were extracted was reduced as a result of this procedure. This aids in the differentiation of various forms of finger flexion. An accuracy of 80–92% (full resolution) was achieved in the first and third quarters of 10 separate arm trials. Using the suggested feature extraction approach in conjunction with discrete wavelet transform, they demonstrated that classification accuracy may be improved by as much as 87% by averaging four radio frequency signals. According to the findings of their research, reduced resolutions may achieve high accuracy levels that are comparable to those of full resolution. Furthermore, they carried out pilot research employing a multichannel single-element ultrasound system, using flexible wearable ultrasonic sensors (WUSs) that utilize non-focused ultrasound. Three WUSs were connected to one subject’s forearm, and ultrasonic RF signals were recorded while the person flexed his or her fingers individually. Using WUS sensors, the researchers discovered that they could accurately categorize finger movement with an accuracy of about 98%, with F1 scores ranging between 95% and 98%.

Li et al. [39] collected the muscle activities of participants by using M-mode ultrasound. The data acquired were utilized to train SVM and BP ANNs, which were then used to categorize the movements of the wrist and hands. The SVM classifier had an average classification accuracy (CA) of 98.83% for M-mode and 98.77% for B-mode across the eight subjects’ 13 movements. Regarding the BP classifier, the average CA of M-mode and B-mode was around 98.7 ± 0.99% and 98.76 ± 0.91%, respectively, according to the results. CAs did not vary between M-mode and B-mode (*p* > 0.05). Aside from that, M-mode seems to have potential dominance in feature analysis. Their findings indicate that M-mode ultrasonography may be used to detect wrist and finger motions, in addition to other applications. The results of their study also show that M-mode ultrasound can be used in HMI.

Table 1 presents a summary of the different machine learning algorithms, feature extraction methods, and modes of ultrasound devices used to classify different types of finger movements and hand gestures since 2006.

## 4. Discussion

This In this paper, we conducted a review of the research works that used sonomyography (SMG) for controlling upper limb prostheses during the last 16 years, since it was first proposed in 2006 [64]. In this technique, different hand gestures can be classified based on the images or signals captured by the US probe to control the prosthesis with multiple degrees of freedom. Because ultrasound imaging can provide information about both superficial and deep muscle activities, this HMI method has a lot of potential for controlling prostheses with more degrees of freedom. To classify hand gestures for controlling robots, various machine learning algorithms and deep learning methods are needed. However, machine learning algorithms are not able to process all the information collected from US transducers. Hence, different transfer learning models have been proposed to extract the characteristics of the collected data and use these features to train the model. The results of this review showed that the most popular algorithms used to categorize the different hand gestures with an accuracy of about 95% from the data collected by US devices are SVM, RBFANN, BP ANN, LDA, K-NN, MLP, SDA, and PCA.

To control a prosthesis using SMG, three different US imaging modes are utilized, namely A-mode, B-mode, and M-mode. The result of this review paper shows that the accuracy of the SMG method with three modes can be good enough to be used for controlling prostheses. The A-mode US system uses very tiny transducers; thus, it can make the system very compact, and US transducers can be integrated with EMG electrodes. However, because the detailed activities of neighboring muscles can be detected in B-mode ultrasound, the reliability of using this US mode may be higher than others. Moreover, the precision can be increased by utilizing different machine learning algorithms in combination with distinct feature extraction methods.

Despite the fact that recent studies have demonstrated the feasibility of using US transducers to control robotic hands, this method has some limitations. Because this method can detect residual muscle activity, it is only appropriate for prostheses in people with a transradial hand amputation level or lower. Moreover, current ultrasound imaging systems are bulky and power-hungry, thus making the prosthesis large and heavy. Furthermore, ultrasound gels or gel pads were used in the published studies for acoustic coupling; thus, the subject’s skin was exposed to moisture for a long period of time, which may have the potential to cause skin problems.

According to this review, the following areas should be further explored and developed for a wider application of SMG for both prothesis control and functional muscle assessment [66]. Firstly, it is necessary to develop a wearable ultrasound imaging system that can be worn by the subject or installed together with their prothesis with dimensions that are sufficiently compact. Recently, a wearable ultrasound data-collection device for muscle functional assessment was demonstrated [63]. Therefore, research works can be focused on further reducing the dimension of the ultrasound system, the footprint of the transducer, and the power consumption of the system, with the battery lasting long enough for the subject’s daily living activities. Wang et al., for example, recently developed a small, lightweight, wireless, and wearable bioadhesive ultrasound (BAUS) device that provides images from organs for 48 h [67]. Secondly, it is very important to solve the acoustic coupling between the skin and ultrasound transducer for practical application of SMG for prothesis control, as the subject may wear their prosthesis for a long time every day. The traditional ultrasound gel or gel pad, which is designed for a short period of use, may not be suitable for this application. In addition, the gel or gel pad coupling may be affected by motion artifacts, thus affecting the performance of SMG control. Recently, it has been demonstrated that some biocompatible materials can serve as a coupling medium for long-term ultrasound imaging of the human body [67]. Similar materials can be used for the future study of SMG prothesis control. Thirdly, all the research works published so far have used a computer to process the ultrasound signal, some in real time and some offline. While it has been demonstrated that real-time signal or image processing, which is required for real-time prosthesis control, is feasible, it requires a high-end computer. For practical use of SMG control for protheses, such signal or image processing must be integrated into a compact and low-power-consuming microprocessor, which should be ultimately installed into the prosthesis for daily activity. Thus, the improvement of efficiency and speed of signal or image-processing algorithm should be an important future direction. Last but not least, SMG provides information about different muscles, and with multiple transducers arranged at various locations, we are able to collect images of muscles involved in complicated hand actions. Thus, it is possible to provide more degrees of freedom for prosthesis control, using more advanced algorithms, such as various deep learning methods. Additionally, SMG control can be combined with the sEMG technique in order to classify more complicated hand gestures with high accuracy, using their complementary features. For instance, the finger movements can be classified using the B-mode SMG signals; meanwhile, to detect the intended wrist movement, including rotation, flexion, and extension, the sEMG signals can be combined.

## 5. Conclusions

According to the review of SMG conducted in this paper, we conclude that SMG has great potential as a novel HMI method for controlling prostheses. It has been clearly demonstrated that SMG signals collected in A-mode, B-mode, and M-mode ultrasound imaging can be used for controlling prostheses effectively. Various machine learning methods have been successfully used to extract control signals from SMG to control prostheses with multiple degrees of freedom by classifying different hand gestures and finger movements. SMG for prosthesis control is becoming a more mature technique since it was first proposed in 2006. Since ultrasound can inherently detect both deep and superficial muscle movements, as well as neighboring muscle activities, SMG has great potential for controlling advanced prostheses with multiple degrees of freedom. With the further improvement of SMG systems by reducing the dimension and cost and increasing the accuracy and battery life, and solving the acoustic coupling issue, SMG has the potential to become a popular HMI method in the future.

## Figures and Tables

**Figure 1 sensors-23-01885-f001:**
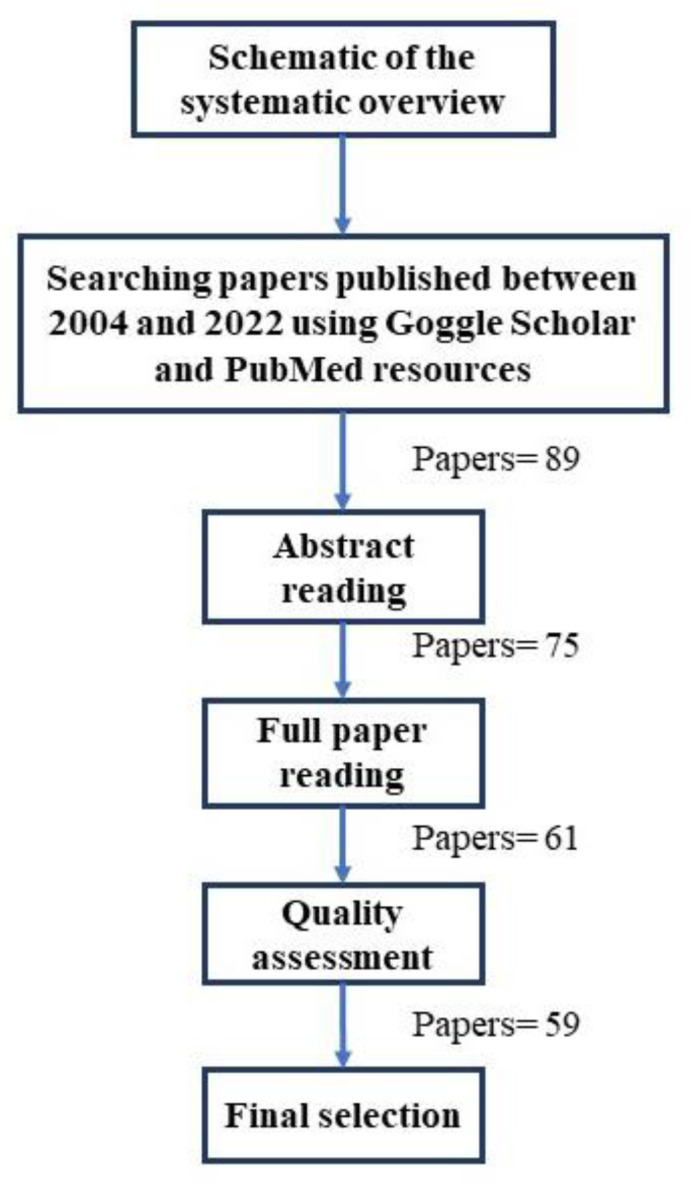
An illustration in schematic form of the overall systematic overview.

**Figure 3 sensors-23-01885-f003:**
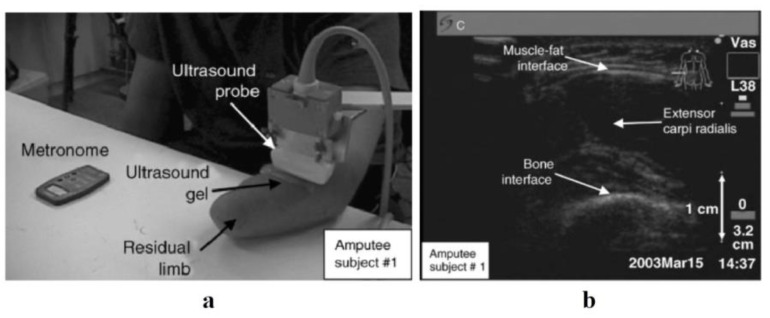
Collecting SMG signals from an amputee subject, using a B-mode SMG system [38]. (**a**) The experimental setup. (**b**) A typical B-mode image of the residual limb. A study by Shi et al. [56] analyzed the possibility of real-time control of a prosthetic hand with one degree of freedom, utilizing muscle thickness fluctuations recorded by a US probe. They investigated the feasibility of controlling a prosthetic hand, utilizing the extensor carpi radialis thickness deformation, and found that a 1-DOF prosthetic hand can be controlled by only one muscle of the forearm, using the SMG technique.

**Figure 4 sensors-23-01885-f004:**
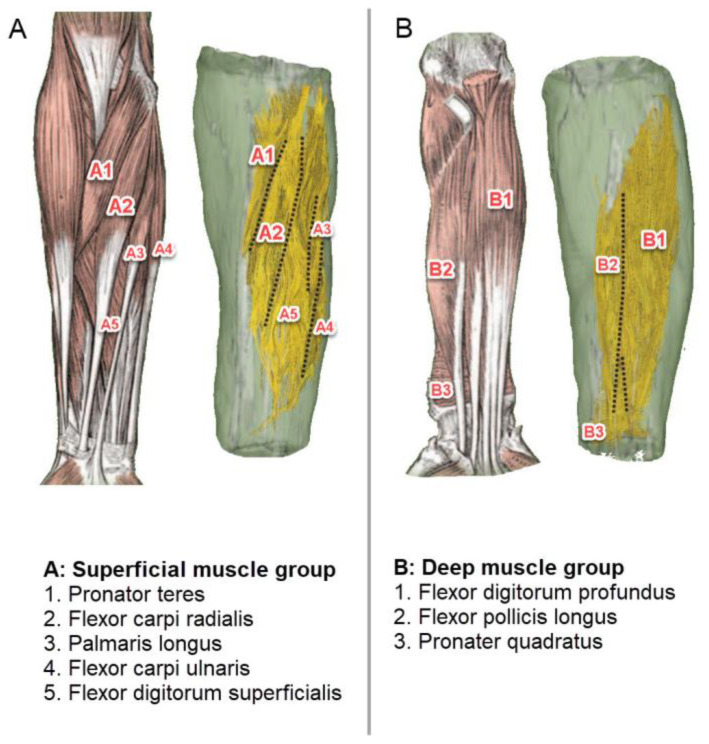
Illustration of fiber tractography and textbook anatomical structure of main forearm flexor muscles [62]. (**A**) Superficial muscle groups of a forearm. (**B**) Deep muscle group of a forearm.

**Figure 5 sensors-23-01885-f005:**
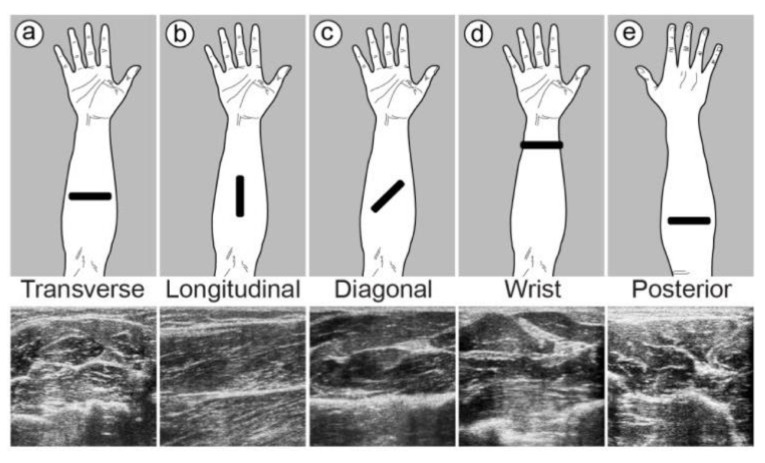
(**a**–**e**) A comparison of the ultrasound probe’s various hand mounting positions, along with the related picture [59].

**Table 1 sensors-23-01885-t001:** Summary of the methods and results of the SMG controlling system used in the past 16 years.

Authors	Year	Ultrasound Mode	Feature Extraction Method	Machine Learning Algorithm	Subjects	Location	Targeted Muscles	Probe Mounting Position	Fixation Methods	Results
Zheng et al. [38]	2006	B-Mode	N/A	N/A	6 healthy and 3 amputee volunteers	Forearm	ECR	Posterior	N/A	The normal participants had a ratio of 7.2 ± 3.7% between wrist angle and forearm-muscle percentage distortion. This ratio exhibited an intraclass correlation coefficient (ICC) of 0.868 between the three times it was tested.
Guo et al. [47]	2008	A-Mode	N/A	N/A	9 healthy participants	Forearm	ECR	NA	Custom-maid holder	A mean correlation value of r = 0.91 for nine individuals was found based on the findings of a linear regression study linking muscle deformation to wrist extension angle. A correlation between wrist angle and muscle distortion was also investigated. The total mean ratio of deformation to angle was 0.130%/°.
Guo et al. [48]	2009	A-Mode	N/A	N/A	16 healthy right-handed participants	Forearm	ECR	NA	Custom-designed holder	The root-mean-square tracking errors between SMG and EMG were measured, and the results showed that the SMG had a lower error in comparison with EMG. The mean RMS tracking error of SMG and EMG under three different waveform patterns ranged between 17 and 18.9 and between 24.7 and 30.3, respectively.
Chen et al. [50]	2010	A-Mode	N/A	N/A	9 right-handed healthy individuals	Forearm	ECR	NA	Custom-designed holder	SMG control’s mean RMS tracking errors were 12.8% and 3.2%, and 14.8% and 4.6% for sinusoid and square tracks, respectively, at various movement speeds.
Shi et al. [56]	2010	B-Mode	N/A	N/A	7 healthy participants	Forearm	ECR	NA	Custom-made bracket	There was excellent execution efficiency for the TDL algorithm, with and without streaming single-instruction multiple-data extensions, with a mean correlation coefficient of about 0.99. In this technique, the mean standard root-mean-square error was less than 0.75%, and the mean relative root mean square was less than 8.0% when compared to the cross-correlation algorithm baseline.
Shi et al. [57]	2012	B-Mode	Deformation field generated by the demons algorithm	SVM	6 healthy volunteers	Forearm	ECU, EDM, ED, and EPL	Posterior	Custom-maid holder	A mean F value of 0.94 ± 0.02 indicates a high degree of accuracy and dependability for the proposed approach, which classifies finger flexion movements with an average accuracy of roughly 94%, with the best accuracy for the thumb (97%) and the lowest accuracy for the ring finger (92%).
Guo et al. [51]	2013	A-Mode	N/A	SVM, RBFANN and BP ANN	9 healthy volunteers	Forearm	ECR	NA	N/A	The SVM algorithm, with a CC of around 0.98 and an RMSE of around 13%, had excellent potential in the prediction of wrist angle in comparison with the RBFANN and BP ANN.
Ortenzi et al. [21]	2015	B-Mode	Regions of Interest gradients and HOG	LDA, Naive Bayes classifier and Decision Trees	3 able bodied volunteers	Forearm	Extrinsic forearm muscles	Transverse	Custom-made plastic cradle	The LDA classifier had the highest accuracy and could categorize 10 postures/grasps with 80% success. It could also classify the functional grasps with varied degrees of grip force with an accuracy of 60%.
Akhlaghi et al. [58]	2015	B-Mode	Customized image processing	Nearest Neighbor	6 healthy volunteers	Forearm	FDS, FDP, and FPL	Transverse	Custom-designed cuff	In offline classification, 15 different hand motions with an accuracy of around 91.2% were categorized. However, in real-time control of a virtual prosthetic hand, the accuracy of classification was 92%.
McIntosh et al. [59]	2017	B-Mode	Optical flow	MLP and SVM	2 healthy volunteers	Wrist and Forearm	FCR, FDS, FPL, FDP, and FCU	Transverse, longitudinal, and diagonal wrist and posterior	3D-printed fixture	Both machine learning algorithms could classify 10 discrete hand gestures with an accuracy of more than 98%. In contrast to SVM, MLP had a minor advantage.
Yang et al. [52]	2018	A-Mode	Segmentation and linear fitting	LDA and SVM	Eight healthy participants	Forearm	FDP, FPL, EDC, EPL, and flexor digitorum sublimis	NA	Custom-made armband	Finger movements were classified with an accuracy of around 98%.
Akhlaghi et al. [60]	2019	B-Mode	N/A	Nearest Neighbor	5 able-bodied subjects	Forearm	FDS, FDP, and FPL	Transverse	Custom-designed cuff	The 5 different hand gestures were categorized with an accuracy of 94.6% with 128 scanlines and 94.5% with 4 scanlines that were evenly spaced.
Yang et al. [54]	2020	A-Mode	Random Forest technique with the help of the Tree Bagger function	SDA and PCA	8 healthy volunteers	Forearm	FCU, FCR, FDP, FDS, FPL, APL, EPL, EPB, ECU, ECR, and ECD	NA	Customized armband	The finger motions and wrist rotation simultaneously using the SDA machine learning algorithm were classified with an accuracy of around 99.89% and 95.2%, respectively.
Engdahl et al. [55]	2020	A-Mode	N/A	N/A	5 healthy participants	Forearm	NA	NA	Custom-made wearable band	Nine different finger movements with an accuracy of around 95% were classified.
Fernandes et al. [61]	2021	B-Mode	DWT and LR	LDA	5 healthy participants	Forearm	NA	Wrist	N/A	Classification accuracy ranged from 80% to 92% at full resolution. However, at low resolution, the accuracy improved to an average of 87% after using the proposed feature extraction method with discrete wavelet transform, which was considered good enough for classification purposes.
Li et al. [39]	2022	M-Mode and B-Mode	Linear fitting approach	SVM and BP ANN	8 healthy participants	Forearm	FCR, FDS, FPL, FDP, ED, EPL, and ECU	Transverse	Custom-made transducer holder	The accuracy of the SVM classifier to classify 13 motions was 98.83 ± 1.03% and 98.77 ± 1.02% for M-mode and B-mode, respectively. However, the accuracy of the BP ANN classifier was 98.70 ± 0.99% for M-mode and 98.76±0.91% for B-mode.

Abbreviations: Not available (N/A); histogram of oriented gradients (HOG); discrete wavelet transform (DWT); linear regression (LR); support vector machine (SVM); radial basis function artificial neural network (RBFANN); back propagation artificial neural network (BP ANN); linear discriminant analysis (LDA); multilayer perceptron (MLP); subclass discriminant analysis (SDA); principal component analysis (PCA); correlation coefficients (CC); root-mean-square error (RMSE); root mean square (RMS); sonomyography (SMG); electromyography (EMG); two-dimensional logarithmic (TDL); extensor carpi radialis (ECR); extensor carpi ulnaris (ECU); extensor digiti minimi (EDM); extensor digitorum (ED); extensor pollicis longus (EPL); flexor digitorum superficialis (FDS); flexor digitorum profundus (FDP); flexor pollicis longus (FPL); extensor digitorum communis (EDC); extensor pollicis longus (EPL); abductor pollicis longus (APL); extensor pollicis brevis (EPB).

## Data Availability

Not available.

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
