# Peer review of "Controlling Upper Limb Prostheses Using Sonomyography (SMG): A Review"

_sensors, 2023, doi:10.3390/s23041885_

Round 1

Reviewer 1 Report

The article describes a review of ultrasound imaging as a relatively novel technique for controlling upper limb prostheses. The review covers different sensing modes, the conditions for data collection, and the data processing techniques mainly used, covering featuring and classification techniques. This information can be relevant for this field, since HMI based on SMG can be considered a potential and alternative method, even with the lacks mentioned in the manuscript. The manuscript is complete, the results are well discussed and it covers enough aspects to be used as a base for future advancements in the field of forearm prosthesis control. I believe that this article would be able to be published in Sensors. 

As minor concerns, I only suggest including details in Figure 3 about the position of each picture, besides the meaning of yellow colors on two images. 

Author Response

Thank you very much indeed for reviewing our paper. Attached please find our response to your comments.

Reviewer 2 Report

The paper performs a very good review about the control of protheses using Sonomyography. The paper is well written and the systematic review is total for the issue adressed. 

The Introduction, Methodology and SMG are well defined and described. In addition, Disccusion and Conclusions are very complete for the scientific field and researchers. 

Only some related works about the control using sEMG and SMG should be introduced and reviewed in the paper. sEMG is currently one of the most suitable source of information for control protheses and there is not some scientific comments in the Discussion about this combination of sources for the control of protheses. 

Author Response

(The authors gave the same response as above.)

Reviewer 3 Report

This manuscript reviews the results of recently published works in the field of sonomyography (SMG) for HMI control of prothesis, showing the important development trend in the field. It is well written and can be considered for publication but the following concerns should be addressed first.

1. A schematic of the systematic overview of this review article will help the readers to better understand the idea.

2. The arrangement of Figure 1 is a bit wire, the authors may consider to improve that.

3. Other than the adopted commercial ultrasound probe for sensing, are there any researches on the development of ultrasonic transducers for SMG? What are the typical structures and advantages/disadvantages?

4. What are the main challenges and research directions of using SMG in prostheses control?

5. Since this is a sensor journal, more discussions on the sensor development will be more attracted to the readers, such as design consideration, performance, etc.

Author Response

(The authors gave the same response as above.)

Round 2

Reviewer 3 Report

The authors have properly addressed my comments, I do not have further comments and suggest it can be accepted now.